# Revisiting the Functional Zoning Concept under Climate Change to Expand the Portfolio of Adaptation Options

Samuel Royer-Tardif [1,2,*], Jürgen Bauhus [3], Frédérik Doyon [4,5], Philippe Nolet [4,5], Nelson Thiffault [5,6] and Isabelle Aubin [2,5]

1 Centre d'Enseignement et de Recherche en Foresterie de Sainte-Foy Inc. (CERFO), 2440 Chemin Sainte-Foy, Québec, QC G1V 1T2, Canada
2 Natural Resources Canada, Canadian Forest Service, Great Lakes Forestry Centre, Sault Ste Marie, ON P6A 2E5, Canada; isabelle.aubin@canada.ca
3 Chair of Silviculture, Faculty of Environment and Natural Resources, University of Freiburg, Tennenbacherstr. 4, 79103 Freiburg, Germany; juergen.bauhus@waldbau.uni-freiburg.de
4 Département des Sciences Naturelles and Institut des Sciences de la Forêt Tempérée (ISFORT), Université du Québec en Outaouais, 58 rue Principale, Ripon, QC J0V 1V0, Canada; Frederik.Doyon@uqo.ca (F.D.); philippe.nolet@uqo.ca (P.N.)
5 Centre for Forest Research, Université du Québec à Montréal, C.P. 8888, succ. Centre-Ville, Montréal, QC H3C 3P8, Canada; nelson.thiffault@canada.ca
6 Natural Resources Canada, Canadian Forest Service, Canadian Wood Fibre Centre, 1055, rue du P.E.P.S., C.P. 10380, succ. Sainte-Foy, QC G1V 4C7, Canada
* Correspondence: s.rtardif@cerfo.qc.ca

**Abstract:** Climate change is threatening our ability to manage forest ecosystems sustainably. Despite strong consensus on the need for a broad portfolio of options to face this challenge, diversified management options have yet to be widely implemented. Inspired by functional zoning, a concept aimed at optimizing biodiversity conservation and wood production in multiple-use forest landscapes, we present a portfolio of management options that intersects management objectives with forest vulnerability to better address the wide range of goals inherent to forest management under climate change. Using this approach, we illustrate how different adaptation options could be implemented when faced with impacts related to climate change and its uncertainty. These options range from establishing ecological reserves in climatic refuges, where self-organizing ecological processes can result in resilient forests, to intensive plantation silviculture that could ensure a stable wood supply in an uncertain future. While adaptation measures in forests that are less vulnerable correspond to the traditional functional zoning management objectives, forests with higher vulnerability might be candidates for transformative measures as they may be more susceptible to abrupt changes in structure and composition. To illustrate how this portfolio of management options could be applied, we present a theoretical case study for the eastern boreal forest of Canada. Even if these options are supported by solid evidence, their implementation across the landscape may present some challenges and will require good communication among stakeholders and with the public.

**Keywords:** forest vulnerability; adaptive capacity; multiple-use land management; conflicting perspectives; natural processes; high-yield silviculture

## 1. Introduction

Managing forests to sustain ecosystem services in the face of climate change is perhaps the biggest challenge that present-day forestry must overcome. On the one hand, in conjunction with a rising demand for wood and other forest products, there is an urgent need to reconcile forest uses in order to sustain the delivery of ecosystem services by preserving the ecological complexity and inherent resilience of forest ecosystems [1,2]. On the other hand, climate change is altering forest ecosystem integrity and resilience by impacting the

growth, health, and survival of trees and other organisms [3–5], jeopardizing the provisioning of ecosystem services. Those two challenges of sustainable forest management are often addressed independently as interventions either seek to reconcile forest usages or adapt to climate change. We believe that both challenges should be more often integrated together as they share a similar solution—they can be addressed through a portfolio of options.

Anticipating the impact of climate change on forest ecosystems entails many uncertainties, from the future extent of climate change itself to spatial and temporal variability in predicted impacts. In this context of future uncertainties and because "no single solution fits all future challenges" [6], most adaptation frameworks for ecosystem management in the context of climate change advocate for implementing a portfolio of options [6–8]. These frameworks adopt a bet-hedging strategy to distribute the risk of maladaptation and mortality across a landscape by promoting a diversity of community compositions and structures that are likely to respond differently to future conditions. One notable adaptation portfolio is that of Millar et al. [6] who proposed three adaptation options at the stand level: (1) creating resistance by reducing the adverse effects of climate change, (2) promoting resilience, i.e., the capacity of ecosystems to recover from disturbances, and (3) enabling forests to respond to change by facilitating adaptive responses. These three options represent a gradient in the amount of ecological change that is accepted by managers and have now been incorporated into many adaptation frameworks [9–11].

A portfolio of management options is also required to reconcile different forest uses in a given landscape. Indeed, providing for multiple functions is a fundamental aspect of sustainable forest management and requires that, beyond wood production, management strategies consider biodiversity conservation, carbon sequestration, non-timber products, cultural values, and traditional uses, among many other services that forests provide [12]. Some of these services, however, are antagonistic to others such that they cannot be delivered from each individual stand. The conservation of some biodiversity elements, for example, might be threatened by some silviculture activities [13].

In the current context of climate change, it is not only conflicting usages that need to be reconciled but also the different and highly uncertain assumptions that we have on how to best respond to environmental change [14]. While a series of proactive actions have been suggested to manage the adaptation of forest ecosystems [15,16], concerns have been expressed that certain forms of proactive management may reduce the adaptive capacity (see Box 1) of natural ecosystems [17,18]. Similarly, how the resistance, resilience, transition portfolio of options translates into truly diversified options might be limited by the regional context into which it is established and individual perceptions about climate change and adaptation measures. For some forest managers, adaptation should be limited to incremental adjustments to business-as-usual scenarios [19,20], whereas others only envision resistance and resilience options [21]. Given the high degree of uncertainty regarding the success or failure of current adaptation options, we advocate for an expanded risk portfolio that applies different management goals and adaptation options to different portions of a landscape to optimize the overall provisioning of ecosystem services and to spread the risk of failure.

The functional zoning approach was developed to reconcile conflicting objectives, mainly wood production and biodiversity conservation, in the context of forest landscapes managed for multiple uses. It does so by assigning different management objectives to separate parts of the landscape, from no-intervention conservation areas (ecological reserves) to intensively managed, high-yielding plantations [22,23]. This approach could offer a useful framework to help address forest adaptation to climate change, because the different management objectives also provide different adaptation options, and because they offer flexibility to consider the environmental, economic, and societal contexts specific to each forest.

Another core aspect to be addressed in the development of adaptation options is forest ecosystem vulnerability and how it is perceived and addressed by practitioners. Forest vulnerability is the combination of (1) the extent of environmental change a forest

ecosystem is likely to experience (exposure), (2) the degree to which it might be negatively impacted by change (sensitivity), and (3) its ability to cope with the new conditions (adaptive capacity, see Box 1) [24]. Over the last decade, the development of more refined climate and ecological models coupled with increased data availability about species ecophysiology, phenology, and genetics, have led to significant advances in the quantification of forest vulnerability to different climate change stressors [25–27]. Although assessments of vulnerability are highly dependent on the way vulnerability is conceptualized and calculated [28,29], these assessments are routinely used to assess climate change impacts [25,30,31] because they remain the best information currently available on which to base decisions. Moreover, despite uncertainties, one consistent feature that emerges from these assessments is that some forests will be more exposed, sensitive, or adaptive than others, depending on their geographic location, composition, and biotic and abiotic environmental conditions [4,28,32,33]. Forest landscapes are thus heterogeneous mosaics of projected forest responses to climate change [34] and risk levels, and these mosaics change dynamically as each stand within them develops and is exposed to ongoing climate change.

The purpose of this paper is to stimulate discussion on how to reconcile contrasting management objectives and adapt to climate change in the light of forest vulnerabilities. We propose to expand the portfolio of adaptation options by presenting how the different management objectives of functional zoning could apply to different levels of forest vulnerability to define a new portfolio of management options. We first provide an overview of the functional zoning approach and its implementation in different parts of the world. We then detail how varying management and adaptation objectives in different parts of a landscape can accommodate diverse perspectives on adaptation to climate change that stem from the different economic, societal, and ecological contexts through which forests are managed. We then present a theoretical case study for the eastern boreal forest of Canada to better illustrate how we envision bridging functional zoning with climate change adaptation. Finally, we discuss some of the challenges for the implementation of a broader portfolio of options.

**Box 1.** Unravelling the meanings of adaptation.

Adaptation is a broad term that encompasses the inherent capacity of biological or human systems and organizations to cope with and adjust to the consequences of climate change [35,36]. This definition corresponds to the third and least known component of vulnerability to climate change—**adaptive capacity** [24,37]. At the level of species and populations, adaptive capacity may be described as a combination of the evolutionary potential, dispersal ability, life-history traits, and phenotypic plasticity, which are influenced by acclimation, as well as genetic and epigenetic processes [38,39]. At the level of ecosystems, adaptive capacity also comprises ecological memory (the capacity of past ecological states to influence present or future ecosystem response, e.g., via seed banks), cross-scale interactions, ecological functioning, and diversity (including presence of rare species [40]). Beyond ecosystems, the term adaptation also includes the concept of **adaptation actions** intended to reduce the vulnerability of a system in anticipation of climate change [9]. For example, replacing drought sensitive tree species or provenances with more tolerant ones is an adaptation action that reduces the sensitivity of forest plantations to drought [15]. Likewise, adaptation actions may also aim to reduce the vulnerability of forest-dependent economies and societies. Without properly defining the system undergoing adaptation (e.g., species, forest ecosystem, or forest community), these different uses of the word adaptation may be confusing. Throughout this article, we discuss how adaptation actions can be effectively assigned into different management options by relying either on natural adaptive capacity or on active human interventions.

## 2. The Functional Zoning Approach

How to ensure that forests provide a diversity of services is a long-standing debate in forestry. By the end of the last century, foresters began to acknowledge the significant challenges of providing conflicting ecosystem services within a given forest stand—it was proposed to specialize distinct forest areas to fulfill different sets of services while ensuring provision of all desired services at the spatial scale of a forest landscape [41]. Pioneering this thought, Seymour and Hunter [22] proposed a triad functional zoning

approach to reconcile biodiversity conservation with wood production. This approach entailed establishing strict conservation areas free of forest management interventions (corresponding to the International Union for Conservation of Nature protected area categories I and II [42]) to preserve ecological features and processes. To compensate for the loss of timber production in these areas, a portion of the forest landscape was planned to be allocated to high-yield intensively managed plantations. Intermediate to these two contrasting management strategies, the remaining portion of the landscape was planned to be managed under "new forestry principles", i.e., favoring biodiversity and other ecosystem services together with wood provisioning [22]. This division of forests into different management foci can also be seen at a global scale, where intensively managed plantations, protected areas, and multifunctional forests contribute respectively 3, 18, and 81% of the total forest area [43].

Thus far, the functional zoning approach has had some success in real-world landscape management in Canada [23,44], being economically viable, ecologically preferable, and socially acceptable for forest stakeholders and users [23,45,46]. A modelling study in Upper Michigan also suggests this approach to be sustainable in multiple-owner landscapes, despite potential differences in owners' management objectives [47]. In the context of climate change, this approach has been applied to match the level of forest vulnerability in British Columbia (conservation areas under low vulnerability and high-yield plantations at high vulnerability [48]) and was suggested as the most suitable option to sustain species rich temperate forests composed of pines (*Pinus* spp.) and oaks (*Quercus* spp.) in Mexico [49]. Biosphere reserves [50] also represent established and widespread examples of functional zoning that seek to reconcile the conservation of biodiversity with its sustainable use.

## 3. Revisiting the Concept of Functional Zoning in the Context of Climate Change

Adaptation actions to climate change fall along a range of approaches extending from **incremental** to **transformative** measures, according to the level of ecological changes that is accepted or desired [19,51]. For some, maintaining the *status quo* and relying on the inherent adaptive capacity of ecosystems [20] appears to be a suitable option; for others, no intervention is not an option in the face of expected environmental changes [16]. One of the best examples of these different perspectives with regard to adaptation to climate change is found in the debate surrounding assisted migration [52,53]. The active translocation of populations or species to follow the evolution of their suitable climate bears important risks; e.g., economic losses, if translocated individuals fail to establish and die, or ecological risks, if translocated species invade the recipient ecosystem and alter ecological functions. Conversely, based on our anticipation of future climatic conditions and on our best knowledge of ecosystem functioning, taking no action may also bear the risk of losing local tree populations and species with associated impacts on related ecosystems, economies, and societies. As it currently stands, there are many situations where we do not possess adequate information to tell which position is the least risky [53], so that any choice of action or no action relies on the subjective assessment of ecological, societal, and economic risks [52].

The functional zoning approach may help meet the challenge of reconciling these different perspectives, allocating different portions of the landscape to different objectives. In the context of climate change, management objectives of functional zoning can be further developed to provide goals that more specifically address climate change adaptation, as shown in Figure 1.

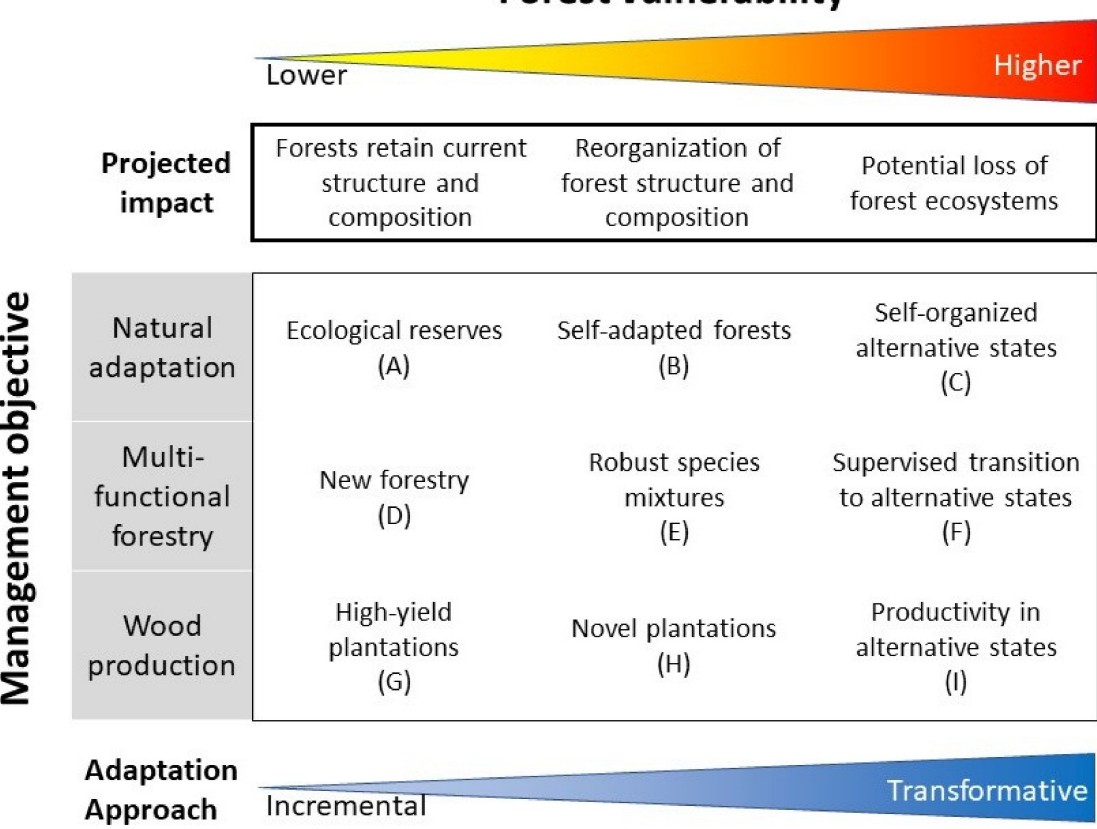

**Figure 1.** The functional zoning approach revisited in the context of climate change to provide a portfolio of adaptation options to climate change combining gradients of management objective and forest vulnerability. The three adaptation measures under lower vulnerability correspond to the traditional functional zoning management objectives [22]. Forests with higher vulnerability are more at risk of undergoing a transition toward alternative non-forest states and therefore transformative measures might be more suited. See the text for a description of management objectives and option names (highlighted in bold in the text), and Table 1 for a short description of each option.

A central element of functional zoning, as originally conceived, was to leave a portion of the landscape devoted to biodiversity conservation (referred to as "**ecological reserve**" by Seymour and Hunter [22]). These areas were to be managed under the assumption that minimizing human interventions in forest ecosystems would maintain natural processes and restore ecological integrity and biodiversity. Such an assumption is undermined by climate change and minimizing human interventions no longer ensures preserving ecological integrity. Nevertheless, promoting natural adaptation becomes one critical management objective under climate change that echoes those preferring to rely on the inherent adaptive capacity of ecosystems to shape adapted forests. Areas originally devoted to biodiversity conservation would be candidates for a **natural adaptation** approach (*sensu* Bolte et al. [54]) that intentionally refrains from active control over adaptive processes [54,55]. In some cases, this approach may have a greater social acceptability than proactive interventions [51], but acceptability may quickly change if climate change threatens the provisioning of ecosystem services or social security, such as, for example, the increasing frequency and intensity of wildland fires [56].

Within the functional zoning approach, the bulk of the landscape is managed to preserve important ecological features (e.g., soil fertility, dead wood, biodiversity) and key ecological processes (e.g., hydrological and biogeochemical fluxes) while ensuring a stable wood supply [57]. Called "**new forestry**" by Seymour and Hunter [22], it integrates up-to-date ecological knowledge of forest functioning to address multiple objectives. In different regions of the world, this family of forest management regimes is referred to

using a range of related terms (e.g., nature oriented, close-to-nature, multi-purpose, and retention forestry), but all correspond to silvicultural alternatives to conventional even-aged forestry [1]. When applied in the context of adaptation to climate change, we refer to this family of management objectives as "**multi-functional forestry**" because it seeks compromises between multiple functions and services while promoting forest adaptive capacity [18]. For example, close-to-nature silviculture, a silvicultural approach widely applied in Central Europe, favors mixed-species forests and aims at retaining the genetic diversity of tree populations by promoting natural regeneration practices that may favor forest adaptive capacity [2,36,58].

The third objective of functional zoning aims at establishing **high-yield plantations** to offset losses of wood/fiber caused by biodiversity conservation in ecological reserves [22]. In the context of climate change, functional zones with **wood production** as their primary management objective may also compensate for potential productivity losses in the zone managed for multi-functional forestry. Moreover, intensifying silvicultural interventions, such as site preparation, vegetation management, thinning, pruning, and fertilization, may accelerate tree growth, increase tree vitality, and shorten the rotation and thus the time period during which a tree crop is exposed to different risk factors. Together with the use of more resistant or faster-growing tree species, productive plantations may reduce the overall risk of tree mortality [59]. As an example of short rotation productive plantations, intensively managed *Pinus radiata* D.Don plantations in New Zealand have a typical rotation length of ca. 28 years [60]. A commitment to wood production as the primary goal of productive plantations would favor the development of precision forestry that seeks the best match between planted genotypes and growing conditions [61]. Plantations with high return on investment may also justify developing novel technologies that increase the resistance and tolerance of commercial tree species to adverse conditions (e.g., drought, pests, and diseases [62,63]). These short rotations also offer frequent opportunities to reset the species and genetic composition of plantations so that it better fits the new environmental conditions (e.g., adapted genotypes [62]) and market needs.

As originally conceived, the three different management objectives of functional zoning (ecological reserves, new forestry, and high-yield plantations) also represent a gradient in management intensity with ecological reserves needing no or minimal interventions, and high-yield plantations requiring much more input to optimize wood or timber production (e.g., site preparation, plantation, vegetation management, fertilization, pruning, thinning, etc.). Conversely, in the context of climate change, our three reformulated management objectives (i.e., natural adaptation, multi-functional forestry, and wood production), as shown in Figure 1, no longer correlate with this gradient of management intensity. The maintenance of specific biodiversity components under climate change might monopolize significant resources (e.g., promoting assisted gene flow between rare populations of *Betula nana* L. in the UK [64]). Conversely, some forests are expected to achieve higher productivity under climate change without additional management interventions because of increased carbon dioxide concentrations and warmer temperatures (e.g., *Eucalyptus* plantations [62]).

**Table 1.** Examples of management options for adaptation to climate change corresponding to different combinations of the three management objectives: favoring natural adaptive capacity (natural adaptation), promoting multiple functions within each stand (multi-functional forestry), and emphasizing forest productivity (wood production) with three levels of forest vulnerability: lower, intermediate, and higher. Letters in the first column refer to Figure 1. Main objective and examples for each option are also provided. [1] New forestry *sensu* [22], i.e., practices that trade single-purpose timber production for a more holistic ecosystem orientation considering stand diversity and biological legacies. [2] Species mixtures that are able to provide services under different environmental conditions. ASCC: Adaptive Silviculture for Climate Change.

| | Objective | Forest Vulnerability | Option | Definition | Example |
|---|---|---|---|---|---|
| A | Natural adaptation | Lower | Ecological reserves | Retain valuable forest features and biodiversity along with their associated cultural values | Biodiversity hotspots in southeastern Canada [65] |
| B | Natural adaptation | Intermediate | Self-adapted forests | Expose local tree populations to climate change and allow natural adaptation | Let-it burn policy in the Sierra Nevada forests to restore fire resilience [7] |
| C | Natural adaptation | Higher | Self-organized alternative states | Allow the natural transition of ecosystems to evaluate the extent of natural adaptive capacity | Northern protected areas acting as refuges while experiencing large ecological changes [66] |
| D | Multi-functional forestry | Lower | New forestry [1] | Reconcile wood production and natural processes based on our best-knowledge of ecosystem functioning | Close-to-nature silviculture in Europe that seeks to favor natural processes guiding ecosystem development [36,58] |
| E | Multi-functional forestry | Intermediate | Robust species mixtures [2] | Ensure forest regeneration and productivity beyond historical ecological boundaries | Cutfoot Experimental Forest resilience trial (ASCC project) [11] |
| F | Multi-functional forestry | Higher | Supervised transition to alternative states | Facilitate transition to new state to retain some of the services provided by forest ecosystems | Converting *Pinus ponderosa* forests to *Juniperus* dominated woodlands to avoid transition to grasslands [67] |
| G | Wood production | Lower | High-yield plantations | Maximize productivity with buffering and resistance measures to shorten stand rotation and increase incomes | *Pinus radiata* plantations in New Zealand [60] |
| H | Wood production | Intermediate | Novel plantations | Sustain wood production by replacing sensitive species with more tolerant ones or promoting species mixtures as an insurance policy | Replacing drought sensitive *Pinus radiata* with *Pinus pinaster* in Western Australia plantations [15] Mixed-species plantations [68] |
| I | Wood production | Higher | Productivity in alternative states | Develop new forest products to maintain productivity and sustain some forest ecosystem services | Mixed *Populus alba–Robinia pseudoacacia* coppice in central Spain [69] |

## 4. From Incremental to Transformative Adaptation Approaches to Address Forest Vulnerabilities

Viewing management objectives through the lens of forest vulnerability may help address the level of management intensity needed and the most efficient way to attain these management objectives. Resistance options devoted to maintaining the *status quo* might be more successful in forests expected to retain their current structure and composition, whereas with increasing vulnerability, managing forests for resilience and eventually transition might be more efficient. Taken differently, adaptation might occur through small incremental actions under low vulnerability, whereas in forests expected to be most impacted by climate change, transformative options might be best suited.

Apart from forest vulnerability, prioritization of actions will depend on the resources available and on a manager's willingness to accept change (itself influenced by its ability to keep meeting social needs). Therefore, instead of viewing management objectives and forest vulnerability as correlated (e.g., intensive wood production occurring only in low vulnerability forests), we present them as two independent gradients that, when intersected, may help expand the portfolio of forest management options, as shown in Figure 1. Our purpose is not prescriptive, but rather intends to present a range of adaptation options within a structured scheme. The nine adaptation options presented in Figure 1, and further detailed in Table 1, should not be taken as discrete fixed options but rather as examples taken along gradients. In this regard, the vertical axis "Management objective" of Figure 1 should be seen as a gradient in the importance given to the promotion of natural adaptation vs. wood production objectives. Similarly, the three projected impacts on the horizontal axis "Forest vulnerability" correspond to examples taken along a continuous gradient of vulnerability and possible outcome. Associated with this gradient of forest vulnerability, we present gradients in adaptation approaches from incremental to transformative measures, as shown in Figure 1.

With increasing levels of vulnerability, efforts may be better invested in transformative measures facilitating the transition towards new states. In order to ensure forest regeneration and productivity under increasing forest vulnerability, management may need to embrace compositions and structures that have no historical analog in the ecosystem to tolerate a wider range of environmental conditions and promote forest resilience. Such actions might be required to ensure that the resulting ecosystems still provide desired goods and services. For example, in the Adaptive Silviculture for Climate Change (ASCC) trial (itself based on Millar et al.'s [6] portfolio of options), transition treatments are based on the assumption that under future climatic and environmental conditions not all tree species of the historical regional species pool are suitable and that this pool should be complemented with new species introduced through enrichment planting [11]. Despite efforts to maintain forest ecosystems, it is possible that highly vulnerable forests will transition to non-forest ecosystems through altered fire regimes, increasing water deficit, or other extreme stress [70]. In some circumstances, a **supervised transition to alternative states** may be necessary to retain some of the services provided by forest ecosystems. In the Southwestern USA, for example, poor regeneration following stand-replacing fires may trigger the conversion of *Pinus ponderosa* C. Lawson forests into shrubland, resulting in net losses of forest ecosystems [71,72]. A transition to smaller trees and shrubs species such as *Juniperus monosperma* (Engelm.) Sarg. and *J. osteosperma* (Torr.) Little, which are better adapted to the anticipated future climate, might be needed to retain some services of forest ecosystems such as carbon storage [67].

As it currently stands, adaptation actions implemented may be biased towards more conservative and less transformative measures. For example, Fischer [19] reported that forest landowner responses in Upper Midwest USA were more incremental (i.e., small changes within the current forest management context) than transformational (large-scale changes that are new to the forest management context). Similarly, ecosystem-based management, an approach that seeks to emulate natural processes, remains a strong component of climate change adaptation in Québec (Canada) public forests [73]. This

may be at least partially attributable to the costs involved with strong transformational changes that require extensive artificial regeneration of new tree species or provenances. Consequently, few transformative options are available for highly vulnerable forests. Here, we advocate that, even in these forests, the three objectives of functional zoning revisited under climate change could be envisioned, as shown in Figure 1. For instance, **productivity in alternative states**, as shown in Figure 1 (Option I), could be secured by very short rotation crops such as willows, poplars, or eucalypts, ideally as mixtures. In central Spain, for example, mixed short-rotation (3 years) coppice composed of 75% *Populus alba* L. and 25% *Robinia pseudoacacia* L. achieved a total yield that was 27 and 90% higher than monocultures of *P. alba* and *R. pseudoacacia*, respectively [69]. In this case, however, the focal production may need to move from industrial roundwood to other purposes (e.g., bioenergy and charcoal) or even non-wood forest products (fruits, nuts, oil, rubber, bamboo, etc.).

In vulnerability hotspots, where ecological changes are highly likely, leaving natural processes to occur may lead to **self-organized alternative states** (Figure 1, option C); an option that is frequently overlooked in adaptation frameworks despite its potential to facilitate adaptation. Disturbances and high mortality rates may accelerate evolutionary processes, particularly for species with a short generation time [74–76]. Disturbances also provide opportunities for the development of novel species assemblages that are adapted to the new environmental conditions generated by climate change [77]. For example, models by Bouchard et al. [78] showed that climate-related tree mortality could offer colonization opportunities for better adapted species and increase northward migration rates provided that landscape connectivity and species migration capacity are sufficient, and that seedlings can properly develop [79]. Consequently, ecological change in unmanaged areas do not entirely preclude meeting biodiversity conservation objectives. This is particularly true for northern protected areas that may experience large ecological changes while acting as refuges for many plant and animal species while they migrate across the landscape [66].

Climate projections show that many protected areas will experience climatic conditions they have never experienced in the past [80]. Where the protection of species and communities in such reserves is more important than the protection of natural processes, some form of active adaptation may be necessary. Conversely, it might also be important for human communities to adapt to the expected ecological changes [81]. Regarding the previous example of highly vulnerable *Pinus* forests of Southwestern USA, evidence suggest that fire is a fundamental component of these ecosystems and the conversion of forests to shrublands or grasslands due to frequent fires could parallel conditions prevailing prior to European settlement [82].

## 5. A Theoretical Case Study in the Boreal Forest of Eastern Canada

To better illustrate how and in which context each management option could be implemented, we present a theoretical case study for the eastern boreal forest in Québec (Canada).

### 5.1. Biogeographic Context

This case study covers an area of publicly-owned land located in the North Shore region of Quebec (48.9–50.6° N, 65.8–69.6° W), Canada, as shown in Figure 2. Most of this area is located within the eastern balsam fir (*Abies balsamea* (L.) Mill.)–white birch (*Betula papyrifera* Marsh.) bioclimatic subdomain, but the north of the area also covers the eastern black spruce (*Picea mariana* (Mill.) BSP)–moss subdomain [83]. Annual temperature averages 2.1 °C, with an average minimum of −13.9 °C in January and an average maximum of 16.2 °C in July [84]. Mean total annual precipitation is 920.6 mm of which 258 mm falls as snow (Godbout meteorological station; 49.32°N, 67.62°W). The fire return interval in this area is relatively long (~300 years [85]) and the main natural disturbances are insect outbreaks, especially spruce budworm (*Choristoneura fumiferana* Clem.) [86], and

windthrow [87]. The topography is irregular with steep slopes, with vertical drops of >200 m between mountain tops and deep valleys.

*5.2. Main Management Issues*

The case study area has been extensively harvested over the last century. In the 1920s, early logging activities were located mainly along the northern shores of the St Lawrence River, extending further north through time and as harvesting operations were mechanized beginning in the 1960s. Past management approaches have altered forest age classes significantly, decreasing the proportion of old-growth forest stands in this area [88]. Furthermore, forest management has increased the proportion of balsam fir in stands relative to historical records [88,89]; this tree species is of lower economic value and is vulnerable to spruce budworm (SBW) outbreaks [90]. Moreover, the irregular topography of the area has led to the development of an extensive road network to access forest sites, fragmenting the forest landscape [87,89]. This area is currently harvested and managed using a multi-functional forestry approach (ecosystem-based management [87]). In addition to landscape fragmentation, increasing rarity of old-growth forests and protecting the threatened woodland caribou (*Rangifer tarandus caribou* Gmelin) are the three main ecological issues of forest management in this area [87]. These issues are interrelated as old-growth forests are important habitats for many components of forest biodiversity [91], including woodland caribou. Because it is considered an umbrella species, protecting habitat for woodland caribou will also protect other threatened wildlife species [92]. Forest fragmentation by roads poses a major threat to woodland caribou by increasing its risk of predation by wolves (*Canis lupus* L.) [93,94]. Moreover, increasing the proportion of regenerating stands within the landscape provides preferred habitat for black bears (*Ursus americanus* Pallas), another woodland caribou predator [95]. Together, these ecological issues underscore the need to preserve large areas of unfragmented old-growth forests [87,89]. Given the considerable trade-off between forest management and maintaining intact old-growth forests, it is unlikely that multi-functional forestry alone can adequately address the aforementioned ecological issues. A functional zoning approach would allow for the establishment of large tracts of forest reserves while maintaining wood production. Climate change, however, is likely to complicate achieving this management goal.

Climate warming and increased carbon dioxide concentrations are expected to increase boreal forest productivity in the short term, but these effects could be transient; this trend is expected to reverse beyond 2 °C warming, mainly because of limitations in water availability [96]. In boreal ecosystems, climate change is also expected to increase the severity and frequency of major disturbances such as fire and insect outbreaks [97]. In our area of interest, models indicate a slightly increased risk of fires by the end of the 21st century [33], while SBW outbreaks could remain an ecosystem-shaping disturbance until 2070 [33], especially given the abundance and prevalence of balsam fir in secondary growth forests. However, outbreaks might decrease in severity by the end of the 21st century as the suitable habitat of SBW shifts northward [98]. Lastly, spatial patterns of vulnerability could be quite variable, stemming from specific sensitivities and adaptive capacity of its constituent species. The most abundant species in this region—jack pine (*Pinus banksiana* Lamb.), black spruce, white spruce (*Picea glauca* (Moench) Voss), and trembling aspen (*Populus tremuloides* Michx.)—should be less sensitive to climate change-related stressors than white birch and balsam fir [26]. Similarly, balsam fir shows lower adaptive capacity than jack pine and the spruces (black and white) to climate change, as characterized by its lower levels of phenotypic plasticity and genetic diversity [99]. Taken together, balsam fir-dominated stands might be more vulnerable to climate change and its related stressors in the future.

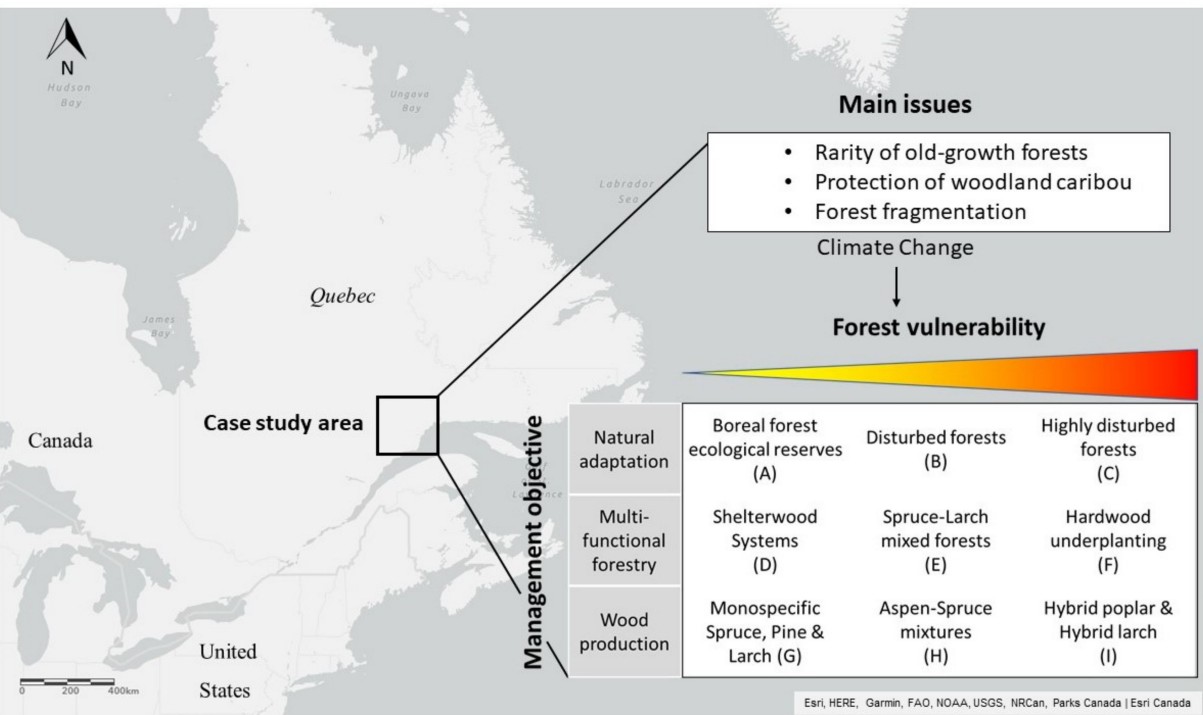

**Figure 2.** Location of the theoretical case study area in Québec (Canada) with the three main forest management issues and an overview of the suggested portfolio of nine adaptation options.

### 5.3. Portfolio of Potential Options

In addition to the rate and intensity of climate change, regional and local factors will also influence boreal forest vulnerability. The case study area receives, on average, more precipitation than boreal regions further west, and its proximity to the Gulf of St Lawrence could buffer climate change to some extent because of its cooling oceanic influence [100]. At the landscape level, topographic shading and temperature inversions in the valleys might provide pockets of localized climate refugia in the boreal forest, at least for the coming decades [100]. Establishing these less vulnerable areas as ecological reserves could be a viable option in order to increase chances of retaining natural and old-growth boreal forests in some areas (Figure 2, option A). More vulnerable areas surrounding these ecological reserves may facilitate natural adaptation and lead to self-adapted boreal forests (Figure 2, option B) or self-organized alternative states, such as parklands or ericaceous heathlands (Figure 2, option C), while retaining elements that are biologically and ecologically beneficial. For example, forest remnants within burned areas may provide suitable calving habitat for caribou [101]. Increasing connectivity between these ecologically important forest patches would reduce forest fragmentation, enabling climate change-related migration of plant and animal species, and facilitate natural adaptation to novel conditions by maintaining gene flow between populations [100,102].

High-yield plantations (Figure 2, option G) could be established in less vulnerable areas to compensate for the decrease in allowable cuts associated with the establishment of large ecological reserves. Native tree species such as black spruce, jack pine, and larch (*Larix laricina* (Du Roi) K. Koch) offer interesting opportunities for high-yield plantations because they respond well to intensive silvicultural treatments [103,104]. For example, scarification, spot fertilization with slow-release fertilizer, and increased foliar N concentration in nursery-raised seedlings can increase the survival and growth rate of black spruce seedlings, even when planted in sites dominated by highly competitive ericaceous shrubs [105]. Moreover, existing local genetic improvement programs for spruces [106,107] and larch [108] can improve yields and survival of native tree plantations [109,110]. Planting a mixture of tree species could further improve plantation diversity and complexity,

while maintaining desirable yields. An example of such novel plantations (Figure 2, option H) is to mix trembling aspen with spruces, recreating a species association that occurs naturally in the boreal forest and enhancing functional diversity all while limiting interference and competition between species [111–113]. Short rotations of high-yield exotic species, such as hybrid poplar (ex: *Populus balsamifera* × *P. maximowiczii*) or hybrid larch (*Larix kaempferi* × *L. decidua*) have proved to be highly productive [114,115]. In vulnerability hotspots, such plantations could serve as options to increase productivity in alternative states (Figure 2, option I).

In the remaining areas, implementing multi-functional forestry practices could be based on recent advances in our understanding of boreal ecosystem dynamics. In the study area, it has been suggested that partial cutting may retain attributes of old-growth forests better than conventional clear-cut systems [87]. Considering this, the use of shelterwood systems and seed-tree treatments could favor the growth and yield of residual trees [116] while ensuring adequate stand regeneration and maintaining the uneven stand structure common to old-growth forests in the region [117]. These systems would represent an example of new silvicultural options in forests with relatively low vulnerability (Figure 2, option D). As an example of robust species mixtures (Figure 2, option E), mixing native black spruce with larch could increase stand diversity while enhancing nutrient cycling and soil fertility. Compared to black spruce, larch produces more litter that is more easily decomposable, promoting increased N cycling and decreasing the C/N ratio of the organic layer [118]. In addition, native larch is also shade-intolerant and has a higher growth rate creating multi-tiered tree canopies in mixed-plantations, thereby possibly decreasing tree competition for light and space [111]. Thinning regimes could also be used in plantations to increase stand resilience to multiple stressors [119], or to diversify stand structure and ecosystem services [120].

Recent hybrid models that integrate tree dispersal scenarios with climate suitability predict that the case study area will become unsuitable for black spruce, balsam fir, and white birch by the end of the century [121]. On the other hand, these models suggest that new hardwood species such as red maple (*Acer rubrum* L.), yellow birch (*Betula alleghaniensis* Britt.), and white pine (*Pinus strobus* L.) could colonize and migrate into this area. In anticipation of this expected shift in dominant species, transition to alternative states could be initiated in some highly vulnerable balsam fir stands to include a greater proportion of these hardwood species (Figure 2, option F). Yellow birch notably has high commercial value on the market and could present an interesting economic opportunity in multi-functional forests.

## 6. Discussion: Some of the Challenges Ahead

We have illustrated how taking into account forest vulnerability in a functional zoning approach may provide opportunities to develop a portfolio of management options for the adaptation to climate change in forested landscapes. Determining how resources should be apportioned among the different adaptation options presented here is highly context dependent and is beyond the scope of this paper. Such attributions will depend on a broad range of factors including the levels of landscape and forest vulnerabilities, on ownership patterns and management objectives, and on managers' perception of risk, predisposition to actively manage forest ecosystems, and willingness to accept change. A diversity of adaptation measures, with some being more transformative, should be implemented to provide a diversification of pathways which could act as an "insurance policy", particularly in large publicly owned lands. In landscapes dominated by small private forests, the diversity of owner perceptions, values, and beliefs may in itself generate a diversity of adaptation options [19,47].

Under climate change, sites that can continue to support productive forests with native species and communities will become increasingly rare and sought after for all adaptation strategies. For example, sites with deep, moist, and nutrient-rich soils may be the most likely to retain large mature trees, an ideal attribute for biological reserves,

and be capable of sustaining tree productivity in the future, a necessary characteristic of productive plantations. Land allocation and land use governance strategies need to be developed to allocate these areas wisely [122]. These lands would be the best candidates for traditional functional zoning objectives, while more transformative measures might be needed in more vulnerable forests [34]. Alternatively, because highly vulnerable forests may be more subject to strong changes in ecosystem structure and composition because of climate change, the sites at which they occur may be more suitable for intensively managed plantations, which bear the risk of altering key ecosystem elements and biodiversity [48].

Social acceptability of transformative measures may be challenging in some parts of the world. In the context of assisted migration, for instance, public controversy may be more likely to arise when tree species are translocated beyond their native range than within [123]. Social acceptability has also been an issue in traditional functional zoning, where intensively managed forest plantations may cause environmental problems judged as unacceptable by the public [68,124]. Improved communication about this form of forestry and its potential benefits is clearly needed [125]. Such benefits include a great potential of sparing lands to be dedicated for conservation if wood production was concentrated in productive plantations occupying a small proportion of forest lands. Indeed, assuming a current demand of industrial roundwood of 2.028 billion $m^3$ and 1.943 billion $m^3$ for wood fuel [126], 265 million ha of moderately productive plantations (15 $m^3$ $ha^{-1}$ $y^{-1}$) would suffice to meet the current global wood demand. This is equivalent to 6.5% of the worlds' forest area of 4.06 billion ha [43]. Developing policies encouraging and guiding the implementation of transformative measures could also help achieve a stronger agreement in the population as well as a better integration of different initiatives within forest landscapes [127].

Climate change will be an increasingly important driver of forest ecosystems. Recent examples from Europe show that the dynamic of forest damage and loss are difficult to predict, partially attributable to the interactions of stress and disturbance agents [128]. Even forests that we currently consider resilient to cope with the climate and disturbances of the coming decades may be pushed beyond their adaptive capacity limits. For this reason, adaptation to climate change is inevitably a continuous adaptive process based on new information and adjustment of objectives.

**Author Contributions:** S.R.-T. and I.A. conceived the idea and wrote the first draft. J.B., F.D., P.N., and N.T. critically reviewed the whole text and provided additional information. S.R.-T., I.A. and N.T. edited the final version. All authors have read and agreed to the published version of the manuscript.

**Funding:** This research was funded by the Forest Change Initiative and the Fibre Solutions Program (Canadian Forest Service, Natural Resources Canada), and the project Forêt s'Adapter itself supported by NSERC (RDCPJ 485153—15), the Consortium on Regional Climatology and Adaptation to Climate Change Ouranos, and the Coop des Frontières.

**Institutional Review Board Statement:** Not applicable.

**Informed Consent Statement:** Not applicable.

**Acknowledgments:** We thank the members of the Forêt s'Adapter project, the Forest Change initiative (Canadian Forest Service, Natural Resources Canada), and colleagues from the Canadian Wood Fibre Centre (Canadian Forest Service, Natural Resources Canada) for constructive discussions that helped developing these ideas. We also thank Françoise Cardou, Laura Boisvert-Marsh, Dan McKenney, Christian Messier, Kim Chapman and John Pedlar for insightful comments on earlier versions of this manuscript. We also thank two anonymous reviewers for providing additional insights on this manuscript.

**Conflicts of Interest:** The authors declare no conflict of interest.

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
