# Peer review of "Revisiting the Functional Zoning Concept under Climate Change to Expand the Portfolio of Adaptation Options"

_forests, doi:10.3390/f12030273_

Round 1
Reviewer 1 Report
Review of the revisions made to Royer-Tardif et al.
The case study improves the manuscript greatly. The logic of the argument is clearer. Although I would not say this is an exceptional paper, it does provide useful ideas that may serve to stimulate advances in the management of forests in the face of climate change.
The phrasing and word choice of the English is not optimal, but it is mostly understandable, so I would rate it as adequate. I leave it to the editor to judge if additional English-language editing is required.
A few editorial suggestions:
L 151. Delete sentence beginning here as being unnecessary.
L 260. Elevated CO2 and longer growing seasons may increase productivity even in the absence of management interventions. OK, I see that you mention this at L 284.
L 356 Typo.
L 426, 460. Revise to: large tracts…
Author Response
We kindly thank the reviewer for the contribution to this manuscript.
We had the manuscript revised for English phrasing and word choice. We hope the new version will better address the reviewer's comment.
Finally, we added all reviewer's editorial suggestions in the new version.
Reviewer 2 Report
Change the title from
Revisiting Functional Zoning Concept Under Climate Change to Expand the Portfolio of Adaptation Options
to
Revisiting the Functional Zoning Concept Under Climate Change to Expand the Portfolio of Adaptation Options
or
Revisiting Functional Zoning Under Climate Change to Expand the Portfolio of Adaptation Options
Author Response
We kindly thank the reviewer for the contribution to this manuscript.
We chose the first title suggestion provided by the reviewer.
This manuscript is a resubmission of an earlier submission. The following is a list of the peer review reports and author responses from that submission.
Round 1
Reviewer 1 Report
Review of “Revisiting functional zoning concept under climate change to expand the portfolio of adaptation options by Royer-Tardif et al. The paper describes a framework to guide the selection of climate-adaptive forest management strategies.
The English is good. The ideas might be helpful to forest managers. However, the organization of the paper makes it difficult to understand what the paper is trying to accomplish, and fails to make the ideas as compelling as they might be. This is driven by a couple of deficiencies in the writing. 1) There is a lot of vagueness around some specific statements. I have pointed out some examples below. 2) The writing is not direct and clear in terms of developing the problem to be solved and generating compelling arguments for the validity and usefulness of the proposed framework. The authors should develop (or better follow an existing) clear outline of logical flow of paragraphs to help the reader quickly grasp the problem and the proposed solution. Within paragraphs, more directness and clarity is needed. 3) The authors state that a discussion of the issues related to deciding how to apply their framework is beyond the scope of the paper (line 338), which may be true. However, I think a case study example would be very helpful for helping readers catch the vision for the framework and demonstrating its usefulness. The paper is not so long that this would be prohibitive, provided that the case study was presented using an “executive summary” approach.
Specific comments.
L39-51. An opening paragraph is key to grabbing the reader with a clear picture of the problem being addressed by the study. This paragraph is too vague. There are several dichotomies presented (ecosystem goods v. services; forest use v. adapting to climate change), but their interactions are not clear, and the problem is not clear.
L52-56. This paragraph seems to flesh out the ideas introduced in the first paragraph, but it too suffers from too much vagueness. I think that this stems from the fact that statements are made, but the reader is expected to look up the citations to find out how and why they are true. I would also introduce your proposed solution in a new paragraph rather than tacking it onto the end of this one.
L69. Where is box 2? Box 1?
L80. Again, the reader is left to look up citations to find out what kind of models the authors have in mind. In fact, this whole paragraph is more vague than it should be because of reliance on unstated content within citations rather than clearly stating the argument. Also, I think this material should be moved to somewhere prior to describing the proposed solution.
L107. Why? Same for sentence in L108.
L133. Sentence starting here is also vague.
L129-147. This paragraph is much clearer than previous ones. However, it is not clear why this information in needed and how it fits into the logic of the discussion. A bit of context would help for this and what follows.
L172. I am aware of a non-Canadian example that you may find useful. Gustafson, E. J., D.E. Lytle, R. Swaty and C. Loehle. 2007. Simulating the cumulative effects of multiple forest management strategies on landscape measures of forest sustainability. Landscape Ecology 22:141-156.
L228. It may be relevant to this paragraph to discuss how climate change may serve to increase productivity through longer growing seasons and CO2 fertilization.
L256. OK, here is a mention of the above. This point does not apply just to plantations. Does your approach consider that plantations may become less needed with overall gains in productivity and reduced demand for some forest products (e.g., paper)? Caution: higher temperatures may help some species in some places, but elsewhere, may subject them to heat stress.
L267. This paragraph finally makes the objective of the paper clear. It’s taken a long time to get to this understanding as the paper is now constructed. This has been obfuscated to this point.
L287. Typo.
L298. This specific example seems out of place in a paragraph making generalities.
L311. Are these gains sustainable over long periods?
L323. Disturbance opens niches, but unless new species can get there and competitively establish, no gains occur. Forest landscape models can address this question. For example, Liang, Y., J.R. Thompson, M.J. Duveneck, E.J. Gustafson, J.M. Serra-Diaz. 2018. How disturbance, competition and dispersal interact to prevent tree range boundaries from keeping pace with climate change. Global Change Biology 24:e335–e351. DOI: 10.1111/gcb.13847.
L354. You should define what you mean by “no regret options.”
L375. Adaptive management comes out of the blue. I suppose the term does not need definition, but its relevance to your framework should be stated somewhere.
Reviewer 2 Report
General comments
This narrative-style paper proposes that functional zoning be considered as a practical means of implementing climate change adaptation across large forest landscapes. This is an old concept relative to forest management in general, but it is a useful contribution to current discussions of climate change. There are no major technical issues here, but there are many small issues, including definitions and jargon, that need to be resolved (see specific comments below).
Although the objectives appear to be mostly conceptual, this paper would be more informative and interesting if there were some illustrations of current and potential use of functional zoning (or similar concepts) in real-world forest landscapes. Some nice examples are provided for individual issues, but there is not much that addresses multiple management objectives across a common landscape. And there is not much that would be of interest to forest managers in a practical sense. It would be helpful to know more about how functional zoning would be enabled by policy and decision making in different landscapes and countries—e.g., public dominated vs. mixed public-industrial-other private lands, coarse-grained spatial mosaic of ownerships vs. fine-grained spatial mosaics. Are there any spatial or temporal considerations or constraints? Functional zoning may be a good idea, but how can it be implemented in a diverse landscape with multiple policies and stakeholders?
It would be relevant to discuss Biosphere Reserves (https://en.unesco.org/biosphere/about) as an established, widespread example of functional zoning. Biosphere Reserves have not always been implemented as intended, but the concept is clear and has international support through UNESCO.
Given the authorship of the paper and the geographic focus of examples and literature cited, it is surprising that key publications on climate change adaptation produced by the Canadian Forest Service (e.g., https://www.nrcan.gc.ca/climate-change/impacts-adaptations/climate-change-impacts-forests/forest-change-adaptation-tools/17770) and Canadian Council of Forest Ministers (e.g., https://www.ccfm.org/english/coreproducts-cc.asp) are not cited. Williamson (CFS) and colleagues, in particular, have provided some of the foundational concepts and literature on climate change adaptation.
Some additional visual appeal would help to illustrate concepts discussed in the paper. Functional zoning is conducive to straightforward maps/diagrams of geographic patterns, thus illustrating on-the-ground examples of implementation (actual or potential). This is an opportunity to demonstrate cases such as the coarse-grain vs. fine-grain spatial mosaics mentioned above.
Specific comments
58-59. This is not just about mitigating negative effects of climate change. Management responses should also include transitioning forest conditions to a warmer climate and/or directing the conditions (species, genotypes, structure, function) to enhance resilience, as mentioned later in the paper.
- “Spread the risk of failure” is an interesting phrase, and I like it. In a more positive vein, one could refer to an “expanded risk portfolio,” per the discussion in the paragraph starting at line 105.
89-92. This sentence has some grammatical problems.
- Landscapes can have “spatial scales,” but they cannot have “levels.” If the term “landscape scale” is used here, then the spatial domain needs to be defined.
113-114. Re “Intersecting it with vulnerability assessments…,” — what does “it” refer to?
- Contractions should not be used.
Table 1. What are “natural adaptation,” “new forestry,” and “close-to-nature silviculture”? These terms are eventually discussed in the text, but they need to be defined when they first appear with the table. In any case, “new forestry” is a vague and dated term that means different things to different people.
- What is “ecological memory”?
139-147. Adaptation can and should also refer to management organizations, not just forests/ecosystems. This can include a wide range of organizational characteristics and behaviors, without which the long-term effectiveness of adaptation on the ground is even more uncertain.
Figure 1. What is a “robust species mixture” in this context? As in Table 1, what are “new forestry” and “natural adaptation”?
- As above, what is “landscape scale”?
187-188. What does “evolution of their climatic niches” mean?
228-229. Re “functional zoning aims to offset timber production losses related to biodiversity conservation with productive plantations,” another perspective is that productive plantations are needed to offset losses of wood/fiber caused by biodiversity conservation.
243-245. This is an important point that is often overlooked.
253-255. This statement is certainly true. However, a more commonly discussed current issue relative to biodiversity in forest management is the revision of genetic guidelines (e.g., seed zones) in forest regeneration. Modifying genotypes of a given species and using different proportions of genotypes in a particular landscape are feasible and have a relatively small risk compared to assisted migration of species.
255-257. Potential growth increases associated with elevated carbon dioxide are uncertain and likely to differ greatly by species and location.
260-266. This is all true but is not novel. Can new insights be provided?
290-299. It should be noted that in some regions, especially the American Southwest, that (1) larger areas burned by wildfires would be similar to conditions prior to European settlement, and (2) conversion of forest to shrubs and grass would parallel what existed during frequent-fire conditions of the past, as well as previous periods of drought (more common prior to the 20th century).
301-304. Management options used by small forest landowners (often with diverse objectives) typically differ from those used by managers of large areas of public lands and industrial forests.
315-317. It seems like “leaving natural processes to occur” is currently the most common approach, right?
- What does “tracking their future climatic niche” mean? This would seem to ignore physiological processes, genetic diversity, and competition.
333ff. This is an effective Conclusions section, although it would be nice to see some ideas on what could be done to accelerate implementation of functional zoning.